# Cessation of breastfeeding in mothers of preterm infants—A mixed method study

**Jenny Ericson** [1,2,3]* , **Lina Palmér** [4]

**1** School of Education, Health and Social Studies, Dalarna University, Falun, Sweden, **2** Center for Clinical Research Dalarna, Uppsala University, Falun, Sweden, **3** Department of Paediatrics, Falu Hospital, Falun, Sweden, **4** Faculty of Caring Science, Work Life and Social Welfare, University of Borås, Borås, Sweden

☯ These authors contributed equally to this work.
* jenny.ericson@regiondalarna.se

## Abstract

### Introduction

Many women cease breastfeeding earlier than desired. This study examined the cessation of breastfeeding among mothers of preterm infants. Thus, the aim was to describe the cessation of breastfeeding in mothers of preterm infants up to 12 months after birth.

### Method

This mixed methods study used a convergent design with both qualitative data, consisting of written comments, and quantitative data, on breastfeeding status and breastfeeding satisfaction. The data were collected from questionnaires sent to the mothers at three points during the first year after birth. In total, 270 mothers of preterm infants who breastfed at the time of discharge from the neonatal unit provided data for the study. The quantitative and qualitative data were analysed separately with statistical tests and hermeneutical analysis, respectively and then together according to the convergent mixed methods design.

### Results

Four themes of the meanings of the cessation of breastfeeding were identified in the qualitative analysis: *"Desire to regain the mother's and the infant's well-being"*, *"The mothers interpretation that the infants actively ceased breastfeeding"*, *"The mother's body and/or the infants' signals showing the way"* and *"The mother's own will and perceived external obstacles"*. Mothers who did *not* breastfeed as long as they wanted were more likely to report less satisfaction with breastfeeding, a shorter breastfeeding period, and less activity when ceasing breastfeeding. In comparison, mothers who breastfed as long as they wanted were more satisfied with breastfeeding, breastfed for a longer period of time and were more active in decision making in breastfeeding cessation.

### Conclusion

Maternal passivity or activity influenced the cessation of breastfeeding in mothers of preterm infants who breastfed at the time of discharge from the neonatal unit. Passive behaviour

**Data Availability Statement:** Data cannot be shared publicly because of ethical and legal regulations. According to the ethical approval, answers from participants should be processed so that unauthorized persons cannot access them. Data are available from the corresponding author

for researchers who meet the criteria for access to confidential data. For that, an ethical approval from the Swedish ethical review authority (www. etikprovningsmyndigheten.se) is needed. An alternative non author point of contact for access to the data underlying the results presented in the study are Dalarna University (www.du.se).

**Funding:** This study was supported by the Centre for Clinical Research Dalarna, Dalarna county, Dalarna University and University of Borås. The funders had no role in study design, data collection and analysis, decision to publish, or preparation of the manuscript.

**Competing interests:** The authors have declared that no competing interests exist.

related to breastfeeding may result in early cessation of breastfeeding, and low breastfeeding satisfaction while active behaviour may increase breastfeeding length and satisfaction.

## Introduction

In Sweden, almost all mothers initiate breastfeeding at birth, but during the first week, 20% cease breastfeeding, and by two months, almost 35% have ceased [1]. In Sweden, there was a decline in exclusive breastfeeding of preterm infants from 2004–2013 [2], and the decline continued from 2013–1017 [3]. Approximately 60% of mothers ceased breastfeeding earlier than they desired. Difficulties with lactation, infant nutrition and weight gain, illness, medication and difficulties with expressing breast milk have been reported to be associated with the earlier cessation of breastfeeding [4].

Mothers of preterm infants are in a vulnerable and fragile situation in which breastfeeding may be considered a key aspect of becoming a mother; in addition, motherhood and breastfeeding often begin in a medical and unfamiliar setting [5]. The initial period of breastfeeding is important since early breastfeeding experiences cause mothers to question their suitability for motherhood [6].Mothers of preterm infants experience breastfeeding in the first 12 months after birth as a journey to finding their own way in breastfeeding, which means that every mother has her own experiences of being in this situation and copes with these experiences according to her own unique situation [7].However, mothers of preterm infants may struggle with breastfeeding for example, with breastfeeding sleepy or immature infants, infant latching, disorganized feeding behaviour or insufficient milk supply, which may continue over a long time [8, 9]. If breastfeeding difficulties occur, a mother may feel threatened and be consumed by concerns about her own body and/or her infant due to pain, discomfort or questions about the amount of milk she is or should be producing. In addition, women's own expectations and/or experiences of objectifying care within the health care system can lead to feelings of loneliness and anxiety [10]. For a mother to have the possibility of breastfeeding as long as she wants, breastfeeding support is crucial [11]. However, mothers of preterm infants have little control over breastfeeding support they receive and inadequate support diminishes breastfeeding [12]. Previous research has found that lower breastfeeding satisfaction, lower self-efficacy, partial breastfeeding at discharge, a low maternal educational level, the use of soothing methods, negative maternal experiences and longer stays in the neonatal unit increased the risk of breastfeeding cessation in mothers of preterm infants [13–15]. In summary, our literature review shows that few studies have examined the cessation of breastfeeding during the first year after birth in mothers of preterm infants; to be able to support breastfeeding, further research in this area is important. The aim of the study is to describe the cessation of breastfeeding in mothers of preterm infants up to 12 months after birth.

## Materials and methods

### Design

The present study adopted a mixed method design with a convergent approach [16]. This design was used to define the relationships between breastfeeding cessation, maternal explanations for breastfeeding cessation, breastfeeding satisfaction and breastfeeding status throughout the first year of life.

### Inclusion and exclusion criteria and setting

During a randomized controlled trial (RCT) conducted after discharge from six neonatal units in Sweden, breastfeeding mothers of preterm infants (gestational age <37 weeks) provided data about breastfeeding cessation during the first 12 months after birth. The results from the RCT are presented elsewhere [17, 18]. The inclusion criteria in the RCT were mothers of preterm infants who breastfed (any breastfeeding) at discharge and had been hospitalized for at least 48 hours in the neonatal unit. Exclusion criteria were mothers who had severe physical or mental illness, language difficulties that could not be resolved, or who had an infant who was transferred to another ward or hospital or where the infant was terminally ill. Eligible mothers were invited to participate in the study approximately one week before discharge. Additional inclusion criteria were providing left written comments on the questionnaire, answering the questions on breastfeeding satisfaction and/or whether the mother breastfeed as long as she wanted. A flowchart over the enrolment is presented in Fig 1. The six neonatal units were level IIIa or IIIb units according to American Academy of Pediatrics Committee on Fetus and Newborn [19]. None of the units were certified as baby friendly. The study received ethical approval from the regional ethical review board in Uppsala, No. 2012/292 and 2012/292/2. After receiving oral and written information about the study, all participating mothers signed a written consent form.

Author JE is a paediatric nurse, and author LP is a midwife by profession with long-term experiences in neonatal and midwifery care. Our preunderstanding and experiences from our professions give us openness to new experiences and insights within the cessation of breastfeeding in mothers of preterm infants. We questioning and continuously reflect over our preunderstanding in relation to analyzing and interpret the data.

### Data collection

Quantitative and qualitative data were collected simultaneously via questionnaires sent to the mothers 8 weeks after discharge from the neonatal unit and 6 and 12 months after the birth of their infants as part of the RCT. The data were collected between March 2013 and December 2015.

Health care professionals collected quantitative demographic data and breastfeeding (exclusive or partial) data at the time of the infant's discharge from the neonatal unit. Breastfeeding (exclusive, partial or no) and breastfeeding satisfaction were measured with self-report questions in the questionnaires at all follow-ups. The World Health Organization's definition of breastfeeding and a 24 hours recall period were used. Exclusive breastfeeding was defined as follows: feeding with breast milk only, regardless of the feeding method, as well as any medications, fortification and vitamins. Partial breastfeeding was defined as follows: feeding with breast milk in combination with formula and/or solid food. No breastfeeding was defined as follows: fully feeding with formula and/or solid food [20]. The questions to both health care professionals and mothers about breastfeeding included the definitions of exclusive, partial and no breastfeeding.

### Measures

Breastfeeding satisfaction was measured with the following question at all follow-ups: "*Are you satisfied with your breastfeeding experience*?" A 10-centimetre visual analogue scale ranging from very dissatisfied to very satisfied was used for responses. Data regarding whether the mother breastfed as long as she wanted were collected in the 12-month questionnaire with the following question: "*If you have ceased breastfeeding, did you breastfeed as long as you wanted*?" The response options were yes or no.

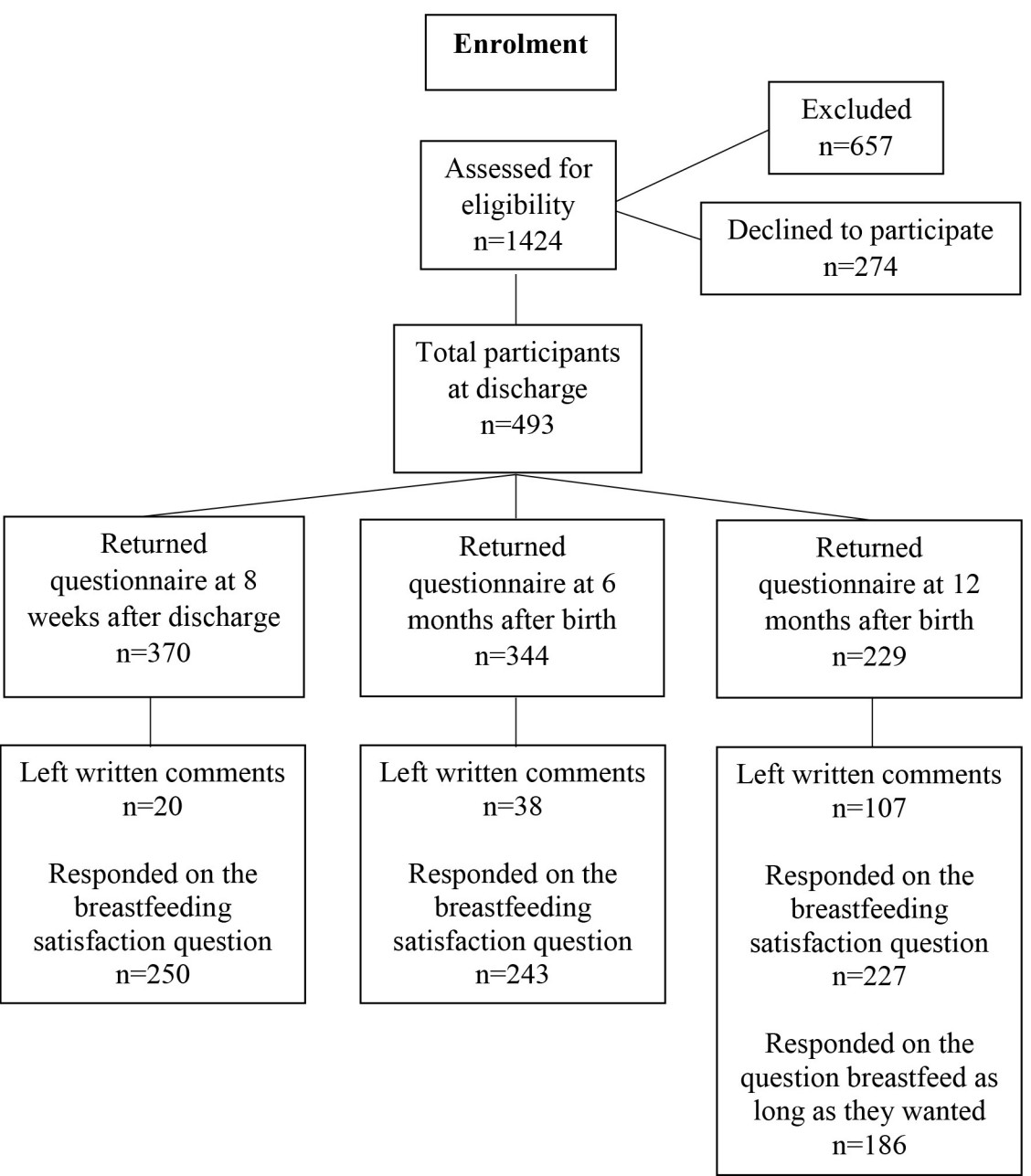

**Fig 1. Flowchart.** A flowchart over the enrolment in the study.

The qualitative data consisted of written comments from the mothers. The comments were collected with one open-ended question (asked at the follow-ups) and one question with a free text option (asked in the 12 month questionnaire). Only the data describing the cessation of breastfeeding were used. In the questionnaires, the following open-ended question was asked at all follow-ups: "*If you want, feel free to write about what you have experienced while breast-feeding/bottle-feeding your baby*". Furthermore, in the 12-month questionnaire, the mothers had the option to provide a free-text response to the following question: "*If you have ceased breastfeeding, did you breastfeed as long as you wanted*?"

## Analysis

The quantitative and qualitative data were analysed separately with statistical tests and hermeneutical analysis, respectively, and then together according to a convergent design, as described by Creswell (2017). In some analyses, the data were divided based on whether the mother breastfed as long as she wanted.

The quantitative data were analysed using IBM SPSS Statistics for Windows, version 25.0 (Armonk, NY: IBM Corp.). The statistical significance level was set to p <0.05. Descriptive statistics were presented as the numbers, percentages, and means and standard deviations (SDs) for normally distributed variables and as the medians and interquartile ranges (IQRs) for non-normally distributed variables. The Mann-Whitney U-test was used to analyse the potential differences between breastfeeding satisfaction and whether the mothers breastfed as long as they wanted. Breastfeeding satisfaction was unevenly distributed. A $chi^2$ test was used to analyse the potential associations between the dichotomous variables i.e., demographic data and whether the mothers breastfed as long as they wanted.

The qualitative data were analysed through hermeneutic analysis based on a reflective life-world approach inspired by hermeneutical and phenomenological philosophy [21, 22]; this analysis aimed to explore the mothers' experiences of breastfeeding cessation. We chose the approach and method in order to provide rigorous scientific foundation for the analysis. The intention of the hermeneutical part of the analysis was to gain understanding of the meanings in the data. The lowest level of the hermeneutical spiral includes the identification of themes related to the meanings in the data, and the most abstract form of explanation is the overall theme. A hermeneutical explanation is not a cause-effect explanation but rather an intentional explanation of the variation in the meanings in the data and why this variation occurs [22].

The mother's written comments were transcribed from the questionnaires to a Microsoft Word document by JE. First, the written comments from the two open-ended questions were read as a whole. Then, the Word document was printed on paper, and all comments regarding the cessation of breastfeeding were cut into separate pieces of paper. These comments were sorted into groups with similar meanings, which ultimately resulted in the identification of four themes of the meaning of breastfeeding cessation. Second, all comments corresponding to each of the four themes were then sorted by whether the mother breastfed as long as she wanted and were marked with each mother's breastfeeding status at each follow-up, resulting in the emergence of a pattern of meaning of breastfeeding cessation. Each comment was marked with the mothers' code, and the same code was used in the quantitative data set. Hence, we were able to connect the written comments with answers to the question about whether the mother breastfed as long as she wanted and breastfeeding outcome. Quotes from the mothers are presented in the results with each mother's randomized code, for example, SU10. Each mother's breastfeeding status at each follow-up, for example, exclusive (e), partial (p) or no (n), is also presented in chronological order as follows: (discharge), (8 w after discharge), (6 months after birth), (12 months after birth). Finally, the total number of months spent breastfeeding is presented, for example, ceased breastfeeding at 8 months (m). Some mothers (n = 16) left written comments in one or more of the follow-up questionnaires but did not answer the question about whether they breastfed as long as they wanted. Their comments did not differ from the other comments; hence, all comments were analysed together.

Finally, an overall theme was interpreted through the linking of the themes of the meaning of the cessation of breastfeeding from the qualitative data and the quantitative data to form an overall theme, i.e., a new whole. To determine the overall theme, an analysis of the mothers' approaches, concepts, words or ways of reasoning, breastfeeding data and breastfeeding satisfaction as well as mothers' descriptions of breastfeeding cessation and its meanings was

**Table 1. Illustration of the analysis and the coding process of the qualitative data.**

| Code | Written comment | Theme of the meaning of the cessation of breastfeeding | The mother breastfed as long as she wanted |
|------|-----------------|-------------------------------------------------------|--------------------------------------------|
| SU40 | I probably would have needed help to cease earlier. I tried for a month and then felt really bad. | Design to regain the mother's and the infant's well-being | No |
| F87 | My daughter started biting me, so I quit. I would have liked to breastfeed for 1 year. | | No |
| F28 | I ceased breastfeeding because of my son's allergy. It was both milk protein and eggs. I was milk-free even when I was breastfeeding. | | Yes |
| Ö54 | My child decided that he wanted to quit:) he was more interested in solid food. | The mother's interpretation that the infants actively ceased breastfeeding | No |
| SU9 | I could breastfeed longer, but my child became uninterested and it became a natural ending | | No |
| T12 | The child did not want to breastfeed anymore, he became uninterested, which made me lose interest too. | | Yes |
| SU11 | In the end, it was only nighttime, and my daughter probably realized that she didn't need it. | | Yes |
| K89 | The breast milk disappeared. | The mother's body and/or the infants' signals showing the way | No |
| T7 | It was lovely to breastfeed; however, my daughter was difficult to feed. She couldn't really breastfeed. In the end, my milk drained and we had to give formula and bottle, which went very well. | | No |
| F64 | My baby was no longer interested in breastfeeding. | | Yes |
| Ö65 | I breastfeed as long as my baby wanted to breastfeed. After about five months, he had tired; there was not enough milk. | | Yes |
| T20 | I chose to start working after 7 months. I had to stop when the [infant's] father would be home. | Mother's own will and perceived external obstacles | No |
| K72 | Hindered because of work. | | No |
| SU53 | Felt like I wanted to quit, and it went great! | | Yes |
| F67 | To make the father-child relationship better and more harmonious, it has made it easier for us to cease breastfeeding. | | Yes |

conducted. We jointly carried out the analysis between the two authors by grouping and discussing the data, individual themes of the meaning of breastfeeding cessation and overall theme until we reached consensus. An illustration of the analysis and the coding process is presented in Table 1.

## Results

The characteristics of the participating mothers are presented in Table 2. In total, 270 mothers contributed data to the study. The open-ended question and the optional free-text response yielded 165 written comments about breastfeeding cessation from 149 mothers. The mothers who breastfed as long as they wanted left 55 comments, and the mothers who did *not* breastfeed as long as they wanted left 94 comments. Sixteen mothers left written comments but did not answer the question about whether they breastfed as long as they wanted.

Significantly more mothers breastfed as long as they wanted than did *not* breastfeed as long as they wanted. Mothers who breastfed as long as they wanted reported significantly higher breastfeeding satisfaction at 8 weeks after discharge and 6 and 12 months after birth than mothers who did *not* breastfeed as long as they wanted (Table 3).

There was a statistically significant difference in exclusive breastfeeding between mothers who breastfed as long as they wanted and mothers who did *not* breastfeed as long as they wanted at discharge and 8 weeks after discharge, but there was not a significant difference in exclusive breastfeeding 6 months after birth or in partial breastfeeding 12 months after birth (Table 3). Mothers who breastfed as long as they wanted breastfed (any breastfeeding) an

**Table 2. Characteristics of the participants.**

| Demographic variables | n (%) median [IQR*] |
|---|---|
| **Maternal variables** | |
| Age, years | 30 [17] |
| Maternal educational level | |
| Higher education | 150 (56) |
| Upper secondary school or less | 120 (44) |
| Primipara | 153 (57) |
| Mothers not born in Sweden | 16 (6) |
| Vaginal birth | 154 (57) |
| Multiple birth | 22 (8) |
| Gestational age at birth, weeks | 34 [3] |
| Exclusive breastfeeding | |
| at discharge | 222 (82) |
| 8 weeks after discharge | 167 (62) |
| 6 months after birth | 72 (27) |
| Partial breastfeeding 12 months after birth | 48 (21)[#] |

Characteristics of the participating mothers (n = 270) and infants (n = 292).

*IQR = interquartile range

[#]Missing data on 42 mothers

average of ten weeks longer than mothers who did *not* breastfeed as long as they wanted (p<0.001) (Table 3).

There were no statistically significant differences between mothers who breastfed as long as they wanted and mothers who did *not* breastfeed as long as they wanted in maternal educational level, parity, gestational week (<32 or >32) or maternal birth country. However, significantly more mothers with twins than mothers with singleton infants did *not* breastfeed as long as they wanted (Table 3).

The comments from the two open-ended questions resulted in the identification of four themes of the meaning of the cessation of breastfeeding, which are described below.

**The mother's body and/or the infant's signals showing the way** was one of the themes that emerged in mothers' descriptions of the cessation of breastfeeding. The mothers described their perceptions that they had a low milk supply, that the breast milk vanished or that there was not enough breast milk for the infant to be satisfied when breastfeeding and/or to gain weight. Some mothers also explained that when they started to give the infant formula or solid food, the breast milk dried up, and it was difficult to continue breastfeeding.

Based on the mothers' descriptions about insufficient milk supply, the drying up of breast milk appeared to happen suddenly, quickly and/or without warning. Negative feedback from the body e.g., insufficient milk supply seemed to reduce the mothers' belief in their ability to breastfeed. The body "lived" its own life and the body thus became an object that the mother adapted to. The mothers became passive and seemed nonplussed.

*The milk started to dry up. Did not have enough milk. SU21, p, p, n, n (ceased breastfeeding at 3 m)*

*he breastmilk vanished. K89, p, p, n, n (ceased breastfeeding at 3 m)Did not have enough breast milk; in the end, my daughter would rather have the bottle. F107, p, p, n, n (ceased breastfeeding at 4 m)*

**Table 3. Breastfeeding satisfaction, exclusive breastfeeding and demographic factors for mothers who breastfed as long as they wanted and mothers who did *not* breastfeed as long as they wanted (n = 185).** Presented as the percentage (%) or median [interquartile range, IQR] and p-value.

| Breastfed as long as they wanted | Yes | No | p-value |
|---|---|---|---|
| Breastfed as long as they wanted | 107 (57) | 78 (43) | <0.001 |
| Maternal educational level | | | 0.35 |
| Higher education | 63 (59) | 49 (63) | |
| Low maternal educational level (upper secondary school or less) | 44 (41) | 29 (37) | |
| Parity* | | | 0.18 |
| Primipara | 59 (56) | 49 (64) | |
| Multipara | 47 (44) | 28 (36) | |
| Country of birth | | | 0.46 |
| Mothers born in Sweden | 103 (96.3) | 74 (94.9) | |
| Mothers not born in Sweden | 4 (3.7) | 4 (5.1) | |
| Multiple birth | | | 0.04 |
| Singleton | 101 (94.4) | 67 (86) | |
| Twins | 6 (5.6) | 11 (14) | |
| Gestational age at birth | | | 0.10 |
| <32 weeks | 10 (9.3) | 13 (17) | |
| 32–36 weeks | 97 (90.7) | 65 (83) | |
| Breastfeeding satisfaction | | | |
| 8 weeks after discharge | 9.2 [2.2] | 7.4 [4.6] | 0.002 |
| 6 months after birth | 9.3 [2.4] | 7.7 [5.2] | 0.003 |
| 12 months after birth | 9.0 [2.5] | 6.8 [5.9] | <0.001 |
| Breastfeeding at discharge | | | 0,001 |
| Exclusive/partial | 9 (8.4) | 21 (27) | |
| No | 98 (91.6) | 57 (73) | |
| Breastfeeding 8 weeks after discharge | | | 0.42 |
| Exclusive/partial | 5 (4.7) | 10 (13) | |
| No | 102 (95.3) | 68 (87) | |
| Breastfeeding 6 months after birth | | | <0,001 |
| Exclusive/partial | 18 (17) | 38 (49) | |
| No | 89 (83) | 40 (51) | |
| Breastfeeding 12 months after birth | | | |
| Partial | 5 (4.7) | 2 (2.6) | 0.37 |
| No | 102 (95,3) | 76 (97.4) | |
| Breastfeeding length, weeks | 35 [17] | 25.5 [24] | <0.001 |

* Missing data on 2 mothers

Regarding this theme, there was a difference between mothers who did and did *not* breastfeed as long as they wanted; mothers who did *not* breastfeed as long as they wanted wrote more descriptions (n = 33) and seemed to cease breastfeeding earlier than the mothers who breastfed as long as they wanted (n = 16). More mothers who did *not* breastfeed as long as they wanted were partially breastfeeding at discharge than mothers who breastfed as long as they wanted. Some of the mothers who did *not* breastfeed as long as they wanted also described their infants as not having the energy/ability to breastfeed. Among the mothers who breastfed as long as they wanted, the most prominent change in breastfeeding was between nine and 12 months after birth, compared to one to six months after birth for the mothers who did *not* breastfeed as long as they wanted.

*The milk [breast milk] was not enough for the infant to be satisfied, and when the bottle came into the picture, the infant did not suck as well on the breast. T1, e, e, n, n (ceased breastfeeding at 6 m)*

Another theme that emerged was **the mother's interpretation that the infants actively ceased breastfeeding**. The mothers explained that their infants were no longer interested in breastfeeding or that the infant did not want to breastfeed. Other descriptions noted that the infant chose to cease or was ready to cease breastfeeding. The mother's interpretation that the infant rejected breastfeeding was a more active action than the mother's interpretation that her body or her infant was showing the way. Perceived negative feedback from the child further reduced the desire to breastfeed among mothers who breastfed as long as they wanted.

*The child did not want to breastfeed; he simply became uninterested, which made me lose my interest. T12, e, p, p, n (ceased breastfeeding at 9 m)*

However, mothers who did *not* breastfeed as long as they wanted expressed a desire to continue breastfeeding.

*My son suddenly chose to stop breastfeeding. I had wanted to continue for a few more months. T5, p, p, p, n (ceased breastfeeding at 10 m)*

Mothers whose data supported this theme breastfed their infants 6 to 13 months, with most of them breastfeeding approximately 9–12 months; however, more mothers who did *not* breastfeed as long as they wanted breastfed for a shorter period.

Regarding the theme of **the desire to regain the mother's and infant's well-being,** the mothers stated that they ceased breastfeeding because of pain; for most of them, this pain was a result of the infant' biting on the breast. Several mothers described their own mental health and medication as reasons for ceasing breastfeeding. In addition, in some cases, the mothers reported that their infants health, such as fussiness and screaming that was associated with an allergy to the protein in cow's milk or other sicknesses. Therefore, ceasing breastfeeding was something the mother did to improve her own or her infant's well-being.

*I wanted to breastfeed for longer, but decreased mental health made the decision to cease breastfeeding the best for everyone. SK50, e, e, p, n (ceased breastfeeding at 8 m)*

A few mothers described that a new pregnancy hindered the continuation of breastfeeding; all of these mothers breastfed as long as they wanted. The mothers described that their breasts were sore, they felt unwell, their milk supply decreased or the infant did not breastfeed because of a new pregnancy. One mother wanted to cease breastfeeding to regain menstruation to become pregnant again.

Most mothers in this theme, including both mothers who breastfed as long as they wanted and mothers who did *not* breastfeed as long as they wanted, breastfed for 9–12 months. The exceptions were mothers who indicated their own mental health as the reason for breastfeeding cessation; these mothers ceased early (1–3 months).

**The mother's own will and perceived external obstacles** were additional reasons to cease breastfeeding. An external obstacle was returning to work. Mothers described returning to work either as obstacle or as a choice; work was described as an obstacle only by mothers who did *not* breastfeed as long as they wanted. One mother wrote that breastfeeding did not work at all, while others wrote that they wanted to breastfeed for longer.

*I chose to start working after seven months. I was forced to cease breastfeeding when [infant's] the father would be home. T20, e, e, e, n (ceased breastfeeding at 7 m)*

Several mothers wrote that they wanted to cease breastfeeding or that they felt that they had breastfed enough; however, these feelings were described only by mothers who breastfed for as long as they wanted. Some mothers described that they ceased breastfeeding to get more sleep at night, while other mothers ceased breastfeeding so that they could share the feeding with the father.

*I felt that I wanted to cease and it went great. SU 53, e, e, p, n (ceased breastfeeding at 8 m)*

The mothers who breastfed as long as they wanted breastfed for eight to >12 months or more, while the mothers who did *not* breastfeed as long as they wanted breastfed for 2–12 months.

## Overall theme: Breastfeeding cessation–an act based on passivity or activity

The triangulation and interpretation of the qualitative and quantitative data revealed that the mothers who breastfed as long as they wanted and the mothers who did *not* breastfeed as long as they wanted showed some similarities. However, they also differed in terms of the meaning of breastfeeding cessation and how many mothers described the reasons for cessation.

In the analysis, it was observed that the mothers who did *not* breastfeed as long as they wanted were less active in promoting their breast milk supply and were less active when breastfeeding ceased. These mothers also described less harmonious breastfeeding and used powerless language when discussing the cessation of breastfeeding. They described their experiences of cessation with phrases such as "dried up", "not enough", "ran out", "was not enough", "unfortunately, [the milk] left", "never got [the milk]", "there is nothing" or "nothing comes". Such expressions were interpreted as indication their passivity and their not taking control over their bodies and milk production, which were related to thoughts of having a biologically predetermined amount of breast milk that could not be influenced by the mother herself, even if the infant was breastfeeding. In other words, these mothers saw their lack of breast milk as something they could not do anything and therefore passively accepted it. They were also generally more dissatisfied with breastfeeding and breastfed for a significantly shorter time than mothers who breastfed as long as they wanted. If the mother took a passive approach to her body's ability for milk production, there was a risk of her being more passive in breastfeeding and a risk of breastfeeding cessation before she wanted.

On the other hand, mothers who breastfed as long as they wanted seemed to be more active in making decisions and to have power over breastfeeding cessation. These mothers also described responses from their infant that they interpreted to indicate that the infant did not want what was offered. However, these mothers described more harmonious breastfeeding (i.e., breastfeeding went smoothly with no major problems or difficulties) and used more empowered language when discussing the cessation of breastfeeding. For example, when they described their experiences, they used phrases such as "decided", "chose", "lost interest", "feel ready", "does not want" or "satisfied". The meanings of such language suggest that the mothers perceived their own body's ability to be more powerful and influential than did mothers who did not breastfeed as long as they wanted; in addition, they perceived the amount of breast milk to be something that they themselves could control. Therefore, mothers who breastfed as long as they wanted were interpreted as taking an active approach to allow them to take control over the breastfeeding situation and cessation. The mothers who breastfed as long as they wanted were generally more satisfied with their breastfeeding and breastfed significantly

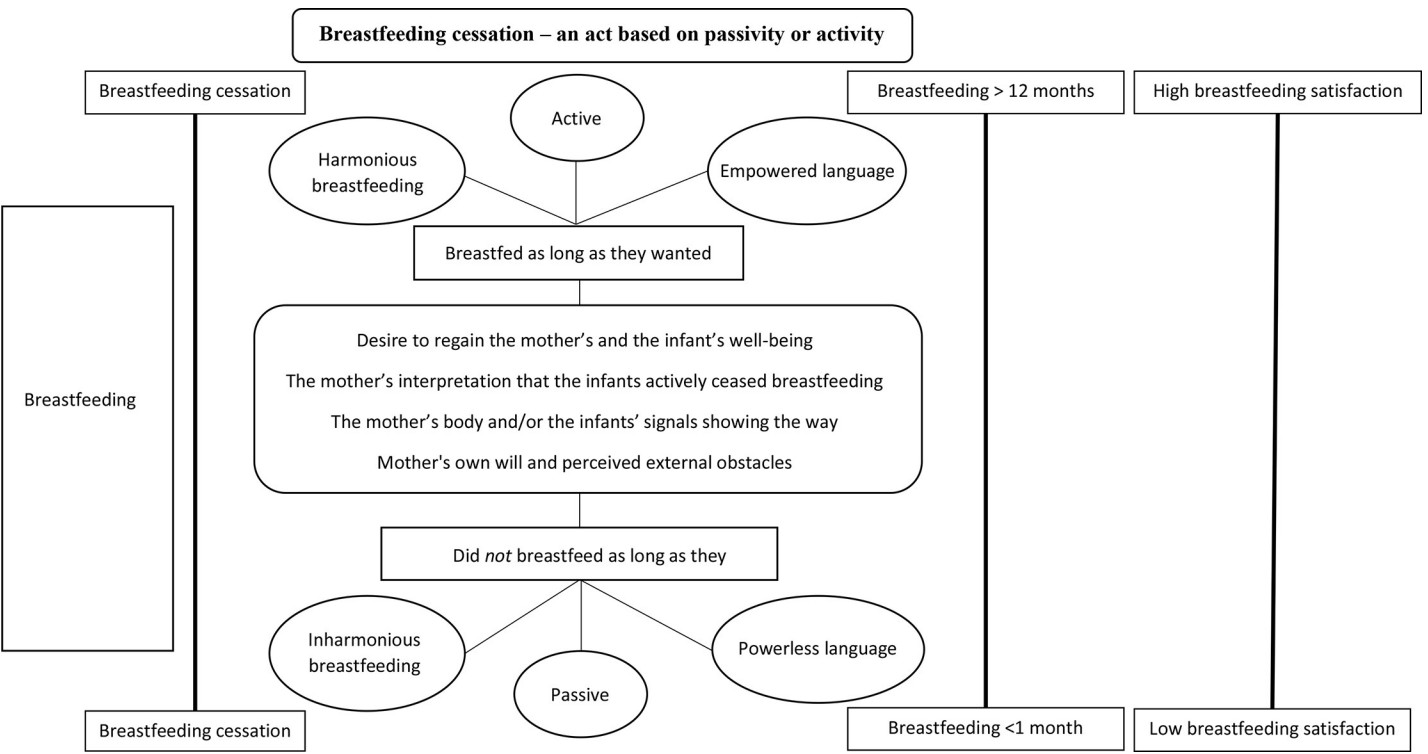

**Fig 2. Schematic figure.** A schematic figure of the overall interpretation and the themes of the meaning of breastfeeding cessation in relation to breastfeeding length, breastfeeding satisfaction and breastfeeding cessation.

longer than the mothers who did not breastfeed as long as they wanted. Being more active in decision making and taking power seemed to facilitate breastfeeding. A schematic figure of the results is shown in Fig 2.

## Discussion

The results of this study showed that for mothers of preterm infants who breastfed at the time of discharge from the neonatal unit, the decision to cease breastfeeding seemed to depend on the mother's passivity or activity in relation to her body's ability, her breast milk production, and her own will as well as the infant's behaviour and signals. Mothers who did *not* breastfeed as long as they wanted were less satisfied with breastfeeding, breastfed for a shorter period and were less active; they did not take control over breastfeeding and were not reflective when ceasing breastfeeding. In contrast, mothers who breastfed as long as they wanted were more satisfied with breastfeeding, breastfed for a longer period of time and were more active in decision making and in taking command in breastfeeding.

The mothers who did *not* breastfeed as long as they wanted breastfed an average of 10 weeks less than did the mothers who breastfed as long as they wanted. This shorter breastfeeding duration may reflect that the mothers had breastfeeding problems. In our study, a few mothers described breastfeeding problems such as mastitis, wounds and/or cracked nipples as reasons for breastfeeding cessation, which has been a common finding in other studies [23, 24]. However, in our study, many mothers described issues with their milk supply. This has also been described in other studies, for example, that of Gianni et al. (2018), who found that mothers who were admitted to a neonatal unit and had problems expressing breast milk or provided an inadequate amount of breast milk had a higher risk of breastfeeding cessation

prior to discharge [25, 26]. Even in studies with mothers of full-term infants, concern about milk supply was a cause of breastfeeding cessation [27, 28]. Additionally, Collin et al. (2002) showed that the infant's behaviour and signs of being unsatisfied were interpreted as an indicator of an insufficient milk supply [28]. In our study, the mothers did not seem to take action to address milk supply issues, such as trying to increase milk production or seeking help. This inaction may have been due to a lack of knowledge or trust in their own ability to breastfeed. Avery et al. (2009) suggested that mothers who are confident about breastfeeding and breast milk production during pregnancy develop a "confident commitment", in which the decision to continue breastfeeding is made [29]. Without such commitment, the cessation of breastfeeding may follow challenges with breastfeeding, which indicates that breastfeeding is a learned skill and not a predetermined skill. This suggestion is interesting in relation to the results of the present study, which indicated that mothers who did *not* breastfeed as long as they wanted were passive and had a predetermined negative view of their body's ability to breastfeed and to produce breast milk. Both Dykes (2006) and Martin (2001) highlighted that the Western view and industrialized dualistic manner of thinking about women's bodies as producers and as objects may have consequences for individual women, who imagines themselves as being alienated from their own bodies [30, 31]. Dykes (2006) suggested that doubt and mistrust towards the body and breast milk production, presented in the present study as passivity and not taking control, can be a consequence of a Western view of women's bodies, which become dominated into being passive objects. Instead, the breastfeeding body must be considered from a non-dualistic way of thinking, and the breastfeeding experience must be seen as an embodied experience [30]. It seems to be important for health care professionals who support mothers of preterm infants in breastfeeding to be aware that mothers seem to have different views on the ability of their bodies to produce breast milk. Mothers who tend to see their bodies as passive objects and who do not take control in an active way must be strengthened to believe in their bodies ability to breastfeed as long as they want. Genuine support strengthens mothers, as shown by Ericson and Palmer (2018). Genuine support is individually adapted and includes both practical and emotional support. Furthermore, genuine support also includes being listened to and being met with respect, understanding and knowledge [12]. In a caring situation in which support is provided, there must be an openness towards each mother's unique breastfeeding situation. Such openness was described by Galvin and Todres (2009) as openheartedness [32]. Being openhearted involves presence for the other person, embodiment and practical responsiveness.

Another interesting result was that many mothers, especially those who breastfed as long as they wanted, stated that their infants wanted to cease breastfeeding and/or lost interest in breastfeeding. Most mothers who breastfed as long as they wanted ceased breastfeeding approximately 8–12 months after birth, which is the time when solid foods are introduced according to national recommendations. In Sweden, few mothers breastfeed their infants after one year of age [1]. This can be compared to a study examining breastfeeding length in non-industrial populations, which showed that breastfeeding until two to four years of age was common [33]. A book on cultural perspectives on breastfeeding claimed that both culture and the medicalisation of breastfeeding are responsible for the shortened periods of breastfeeding in the Western world. Observations of the mother-infant relationship in traditional societies has shown that all mothers breastfed their children, often until 3 to 4 years of age, which is also supported by palaeontological evidence [34].

## Limitations

The trustworthiness of the study is strengthened by the mixed method design, which strengthens the understanding of breastfeeding cessation through the use of both qualitative and

quantitative data [16]. Many mothers provided similar comments on the cessation of breast-feeding; hence, the interpretation of the themes of the meaning of breastfeeding cessation is likely trustworthy. Additionally, the results can probably also be transferred to similar contexts because of the relatively large number of comments and the similarities in the comments on cessation of breastfeeding despite the participants being spread over a large part of Sweden. Although the comments provided by the mothers were relatively short, there were many comments, leading to the identification of a wide variety of meanings in the data instead of an in-depth description of meanings. This wide variety of meanings may be of interest to investigate in more depth in future research [35]. However, to deepen the understanding of breastfeeding cessation, it may be beneficial to perform in-depth interviews with mothers. The authors remained open to the data but also questioned and continuously reflected on the analysis and results, which strengthened the credibility and confirmability of the findings. A further strength of the study was the long follow-up, during which we measured breastfeeding satisfaction repeatedly and assessed the cessation of breastfeeding close to the time of event/experience. It does not appear that the intervention in the RCT study affected the results of this study. The intervention lasted until 14 days after discharge and did not affect breastfeeding [18]. It seems unlikely that the intervention would have affected the cessation of breastfeeding, which usually happened much later, as supported by the analysis.

A limitation is that the questions measuring breastfeeding satisfaction and whether the mother breastfeed as long as she wanted were not validated. However, the breastfeeding satisfaction question had a strong correlation ($r = 0.70–0.74$) with the validated Maternal Breast-feeding Evaluation Scale [36], which also measures breastfeeding satisfaction. That the mothers answered at all follow-ups.

## Conclusion

Passive and active behaviour influence the cessation of breastfeeding in mothers of preterm infants who breastfed at the time of discharge from the neonatal unit. Passive behaviour increases the risk of early breastfeeding cessation and lower breastfeeding satisfaction, while active behaviour increases breastfeeding length and satisfaction. This is important knowledge when supporting breastfeeding and designing interventions to support breastfeeding.

## Acknowledgments

The authors would like to thank all mothers who participated in the study.

## Author Contributions

**Investigation:** Jenny Ericson, Lina Palmér.

**Writing – original draft:** Jenny Ericson, Lina Palmér.

**Writing – review & editing:** Jenny Ericson, Lina Palmér.

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
