## [Decision Letter · Decision Letter 0]

6 Mar 2020

PONE-D-20-00937

Cessation of breastfeeding in mothers of preterm infants – a mixed method study

PLOS ONE

Dear Dr Ericson,

Thank you for submitting your manuscript to PLOS ONE. After careful consideration, we feel that it has merit but does not fully meet PLOS ONE’s publication criteria as it currently stands. Therefore, we invite you to submit a revised version of the manuscript that addresses the points raised during the review process.

Please consider the detailed comments of the two peer reviewers, including the attachment provided by one of the reviewers, that support a recommendation of a major revision.

The reviewer comments will also help to ensure a revised manuscript fully and clearly is in line with PLoS editorial recommendations that qualitative manuscripts include: 1) defined objectives or research questions; 2) description of the sampling strategy, including rationale for the recruitment method, participant inclusion/exclusion criteria and the number of participants recruited; 3) detailed reporting of the data collection procedures; 4) data analysis procedures described in sufficient detail to enable replication; 5) a discussion of potential sources of bias; and 6) a discussion of limitations. PLoS editorial suggests that authors use the COREQ checklist, or other relevant checklists listed by the Equator Network, such as the SRQR, to ensure complete reporting (http://journals.plos.org/plosone/s/submission-guidelines#loc-qualitative-research) to help facilitate this.

We would appreciate receiving your revised manuscript by Apr 20 2020 11:59PM. To enhance the reproducibility of your results, we recommend that if applicable you deposit your laboratory protocols in protocols.io, where a protocol can be assigned its own identifier (DOI) such that it can be cited independently in the future. For instructions see: http://journals.plos.org/plosone/s/submission-guidelines#loc-laboratory-protocols

We look forward to receiving your revised manuscript.

Kind regards,

Joann M. McDermid, MSc, PhD, RDN, FAND

Academic Editor

PLOS ONE

Journal Requirements:

Reviewers' comments:

Reviewer's Responses to Questions

**Comments to the Author**

1. Is the manuscript technically sound, and do the data support the conclusions?

Reviewer #1: Partly

Reviewer #2: Yes

2. Has the statistical analysis been performed appropriately and rigorously? 

Reviewer #1: No

Reviewer #2: I Don't Know

3. Have the authors made all data underlying the findings in their manuscript fully available?

Reviewer #1: No

Reviewer #2: No

4. Is the manuscript presented in an intelligible fashion and written in standard English?

Reviewer #1: No

Reviewer #2: No

5. Review Comments to the Author

Reviewer #1: The inclusion criteria for both mothers and preterm infants are not explicitly stated. Measures to ensure validity and reliability of the data collection instrument used in the quantitative study was not indicated.The measures to ensure trustworthiness of the qualitative aspect of the study were also not made explicit in the manuscript. Some of the percentages do not sum up to 100%. Some of the words used and sentences need to be revised for clarity.

Reviewer #2: Review of: Cessation of breastfeeding in preterm infants – a mixed method study.

Introduction:

The decline in breastfeeding of preterm infants are based on data from 2004-2013 from the Swedish National Quality Register. Are there not any newer data that can be drawn from the Register to evaluate current breastfeeding status?

You refer to a study of breastfeeding problems as experienced by mothers of healthy infants with a GA of > 35 weeks. Several studies report preterm infants to be in increased risk of breastfeeding problems, and one might argue that breastfeeding problems of mature and premature infants might differ, for instance in prevalence, e.g. sleepy infant, not enough milk (which you also discuss later in your manuscript under discussion) Therefore you should also refer to studies of breastfeeding problems as experienced by mothers of preterm infants.

Line 72-73: To have the possibility of breastfeeding as long as one wants, breastfeeding support is crucial. Yes!, but please support this by a reference.

Method:

The methods section needs to be revised. You should consider sub-headings in ‘Materials and methods’ to enhance the reading of the manuscript, e.g. sub-heading: Design, setting and participants, and sub-heading: Data collection, and sub-heading: Measures.

Please state how many mothers were eligible for study in the study period and how many were approached about the study? You could also add a figure illustration the inclusion process and response rates during the study period. Did you have any ethical considerations in regard to recruiting participants; e.g. critical ill infant, mother with mental disease or psychological issues? When were mothers approached at the neonatal unit? At discharge? A week before?

Please define exclusively and partial breastfeeding, e.g. did you follow WHO’s definition? And how was the mothers informed, as it seems as they self-reported breastfeeding status after discharge? Therefore, please state clearly when breastfeeding status were reported by healthcare professionals (at discharge?) and self-reported by mothers (after discharge?). Please elaborate on setting, e.g. were any hospitals ‘babyfriendly’ (BFIH)?, and inclusion criteria, e.g. were only singleton infants recruited? In Table 1 multiple births appears but should be mentioned in ‘Methods’. No exclusion criteria are mentioned? Please elaborate if you did or did not have any considerations in this regard, e.g. language?

Line 104-106 and line 111-113 are unclear. Did you include data (free text) from the question mentioned in line 111-113 in your qualitative analysis if it described breastfeeding cessation?

Line 139-140: How did you sort these comments as you described that they were sorted by breastfeeding as long as the mother wanted or not?

Coding 6, 12 months after birth is according to the infants real birth date and not adjusted to gestational age (GA)? Please state shortly in line 96 if questionnaires were distributed according to real birth date.

Argumentation for choice of method for qualitative analysis could be improved. Please state if you used any software to analyze the qualitative data.

Please add a table illustrating your analysis and coding process, as it will enhance the reader’s ability to assess the quality of your analysis.

Results:

Response rates during the study period are missing, see comment under methods.

Can you add a comment in regard to how the amount of qualitative data (total comments) were distributed in regard to who breastfeed as long as they wanted and who did not breastfeed as long as they wanted (line 131 -132) as you in your analysis state that comments were lacking in some themes from mothers not breastfeeding as long as they wanted (line 189-190).

Line 178: drying up? Reflect upon wording as your quotes reflects mothers’ wording and the text your academic language, e.g. consider ‘insufficient milk supply’?

Line 190: left or lack?

Discussion:

Line 305 and line 307: You say studies but only refer to one study? E.g. in breastfeeding problems there are many studies relevant to add as references.

Line 342-343. Did you collect data in regard to solid food introduction in your study population? Or are there studies describing that solid food are introduced in preterm infants by 8-12 months after birth? That number is not adjusted to GA? As introduction of solid foods are earlier, as reported in term infants.

Line 345-348. Please make your argumentation clear in short writing. , e.g. remove redundant text.

Line 348: Are there any studies you could refer to as to support your hypothesis?

Please elaborate further on strengths and limitations of your study as it by now is hardly described; e.g. what steps did you take to enhance the validity of your study?

As your study is secondary analysis of data derived from a larger intervention study (RCT), you have not stated how it might have or have not affected the results of your study, which is an important issue to address, when reporting the validity of your findings.

You should also add a short description about how you dealt with/used/reflected on your preunderstanding throughout your study (under Methods) as your preunderstanding can create bias and reduce validity. You could as well state if your e.g. profession could create potential bias.

General reflections:

The manuscript should be grammar checked as there are several typos and edited in the use of English, as some wordings could be improved. Several places in the manuscript authors’ names of references could be deleted as seems redundant text and will improve the reading of manuscript, e.g. for example, Feenstra et al and for example Gianni et al. Replace with only the reference in Vancouver style.

6. PLOS authors have the option to publish the peer review history of their article (what does this mean?). If published, this will include your full peer review and any attached files.

Reviewer #1: No

Reviewer #2: No

---

## [Author Response · Author response to Decision Letter 0]

23 Apr 2020

Dear editor and reviewers

Thank you for the opportunity to revise our manuscript and for the comments that have been given to us. In the revised manuscript we have made track changes to the original version. Withdrawn texts were not retained as crossed-out text. Please find below our replies to the comments.

We have addressed the comments from the editor. We have an objective. We have clarified the description of the sampling strategy, participant inclusion/exclusion criteria and the number of participants recruited. We have clarified the reporting of the data collection and data analysis procedures. We have expanded the discussion of potential sources of bias and the discussion of limitations.

We have used the SRQR checklist to ensure complete reporting of the qualitative part of the study. 

Reviewer #1: The inclusion criteria for both mothers and preterm infants are not explicitly stated. 

Reply: Thank you; we have added that information in the method section.

Measures to ensure validity and reliability of the data collection instrument used in the quantitative study was not indicated.

Reply: We have added a sentence about the validity of the measures in the discussion.

The measures to ensure trustworthiness of the qualitative aspect of the study were also not made explicit in the manuscript. Some of the percentages do not sum up to 100%. Some of the words used and sentences need to be revised for clarity.

Reply: We have revised through the whole manuscript.

Reviewer #2: 

Introduction:

The decline in breastfeeding of preterm infants are based on data from 2004-2013 from the Swedish National Quality Register. Are there not any newer data that can be drawn from the Register to evaluate current breastfeeding status?

Reply: There are no newer published data. However, we added a reference from the Swedish Neonatal Quality register (annual report) that show a continued decline in breastfeeding. Unfortunately, the report is in Swedish, but you may see the figures.

You refer to a study of breastfeeding problems as experienced by mothers of healthy infants with a GA of > 35 weeks. Several studies report preterm infants to be in increased risk of breastfeeding problems, and one might argue that breastfeeding problems of mature and premature infants might differ, for instance in prevalence, e.g. sleepy infant, not enough milk (which you also discuss later in your manuscript under discussion) Therefore you should also refer to studies of breastfeeding problems as experienced by mothers of preterm infants.

Reply: We have added a sentence about mothers of preterm infants.

Line 72-73: To have the possibility of breastfeeding as long as one wants, breastfeeding support is crucial. Yes!, but please support this by a reference.

Reply: Thank you, a reference has been added.

Method:

The methods section needs to be revised. You should consider sub-headings in ‘Materials and methods’ to enhance the reading of the manuscript, e.g. sub-heading: Design, setting and participants, and sub-heading: Data collection, and sub-heading: Measures.

Reply: Thank you for your suggestion, we have added subheadings.

Please state how many mothers were eligible for study in the study period and how many were approached about the study? You could also add a figure illustration the inclusion process and response rates during the study period. 

Reply: We have added a flowchart over inclusion and exclusion and response rate, se figure 1.

Did you have any ethical considerations in regard to recruiting participants; e.g. critical ill infant, mother with mental disease or psychological issues? When were mothers approached at the neonatal unit? At discharge? A week before?

Reply: Thank you, we have added that information under the method section.

Please define exclusively and partial breastfeeding, e.g. did you follow WHO’s definition? And how was the mothers informed, as it seems as they self-reported breastfeeding status after discharge? Therefore, please state clearly when breastfeeding status were reported by healthcare professionals (at discharge?) and self-reported by mothers (after discharge?). 

Reply: Thank you, we have added a definition of breastfeeding and who provided the data.

Please elaborate on setting, e.g. were any hospitals ‘babyfriendly’ (BFIH)?, and inclusion criteria, e.g. were only singleton infants recruited? In Table 1 multiple births appears but should be mentioned in ‘Methods’. No exclusion criteria are mentioned? Please elaborate if you did or did not have any considerations in this regard, e.g. language?

Reply: We have added information of the inclusion and exclusion criteria and neonatal units under settings. Both singleton and twin mothers were recruited.

Line 104-106 and line 111-113 are unclear. Did you include data (free text) from the question mentioned in line 111-113 in your qualitative analysis if it described breastfeeding cessation?

Reply: We have revised the paragraph.

Line 139-140: How did you sort these comments as you described that they were sorted by breastfeeding as long as the mother wanted or not?

Reply: Each comment were marked with the mothers’ code and the same code was used in the quantitative data set. Hence, we could connect the codes with the answer on the question if the mother breastfed as long as she wanted or not. We have added that information in the analysis section.

Coding 6, 12 months after birth is according to the infants real birth date and not adjusted to gestational age (GA)? Please state shortly in line 96 if questionnaires were distributed according to real birth date.

Reply: It was the real birth date, not corrected age. This, have been clarified.

Argumentation for choice of method for qualitative analysis could be improved. Please state if you used any software to analyze the qualitative data.

Reply: We have added more about the analyze process in the text see analysis section.

Please add a table illustrating your analysis and coding process, as it will enhance the reader’s ability to assess the quality of your analysis.

Reply: We have added a table of the analysis process, see table 1.

Response rates during the study period are missing, see comment under methods.

Reply: We have added a flowchart over included and excluded mothers and response rates, se figure 1.

Can you add a comment in regard to how the amount of qualitative data (total comments) were distributed in regard to who breastfeed as long as they wanted and who did not breastfeed as long as they wanted (line 131 -132) as you in your analysis state that comments were lacking in some themes from mothers not breastfeeding as long as they wanted (line 189-190).

Reply: Mothers who did not breastfeed as long as they wanted provided more comments in this theme. We have changed the sentence to clarify.

Line 178: drying up? Reflect upon wording as your quotes reflects mothers’ wording and the text your academic language, e.g. consider ‘insufficient milk supply’?

Reply: We have changed the wording.

Line 190: left or lack?

Reply: They provided more comments. We have clarified the sentence. 

Discussion:

Line 305 and line 307: You say studies but only refer to one study? E.g. in breastfeeding problems there are many studies relevant to add as references.

Reply: Thank you, this have been changed.

Line 342-343. Did you collect data in regard to solid food introduction in your study population? Or are there studies describing that solid food are introduced in preterm infants by 8-12 months after birth? That number is not adjusted to GA? As introduction of solid foods are earlier, as reported in term infants.

Reply: We did not collect data on solid food introduction. However, national guidelines recommend introduction of solid foods at 6 months corrected gestational age. We clarified the sentence in the manuscript. 

Line 345-348. Please make your argumentation clear in short writing. , e.g. remove redundant text.

Reply: We have revised the sentence.

Line 348: Are there any studies you could refer to as to support your hypothesis?

Reply: We have revised the paragraph.

Please elaborate further on strengths and limitations of your study as it by now is hardly described; e.g. what steps did you take to enhance the validity of your study?

Reply: We have clarified the limitations and trustworthiness of the study.

As your study is secondary analysis of data derived from a larger intervention study (RCT), you have not stated how it might have or have not affected the results of your study, which is an important issue to address, when reporting the validity of your findings.

Reply: We have added a paragraph about the RCT and our study.

You should also add a short description about how you dealt with/used/reflected on your preunderstanding throughout your study (under Methods) as your preunderstanding can create bias and reduce validity. You could as well state if your e.g. profession could create potential bias.

Reply: We have added a paragraph about our preunderstanding and professions.

General reflections:

The manuscript should be grammar checked as there are several typos and edited in the use of English, as some wordings could be improved. Several places in the manuscript authors’ names of references could be deleted as seems redundant text and will improve the reading of manuscript, e.g. for example, Feenstra et al and for example Gianni et al. Replace with only the reference in Vancouver style.

Reply: We have made a language review by another professional language reviewer, see attached certificate. We have corrected the references.

Line 40: You need to include the method of data analysis in the methods section of the abstract.

Could include the themes identified in the results section of the abstract.

Reply: We have added the analysis methods and identified themes in the abstract.

Introduction: The introduction was supported by relevant literature however there is a need to provide references for statements made in lines 58, 64-65.

Reply: Thank you, we have added references.

Material and methods: The design is appropriate for the study. Though it was stated that 270 mothers were used for the study in the abstract as well as the results section of the manuscript, the inclusion criteria were not explicitly stated in relation to both mother and infant characteristics. For example, category of preterm infants - extremely, preterm, very preterm or moderate to late preterm. The sampling technique was also not stated. Measures used to ensure validity and reliability of the items in data collection instrument for the quantitative part of the study as well as those used to ensure trustworthiness in the qualitative part should be included.

Reply: We have added information on inclusion and exclusion criteria in the method section. We included breastfeeding mothers of all preterm infants < 37 weeks. We sampled data via questionnaires, which is stated in the method section. The method section have been clarified.

Line 87-89: The statement is not necessary. 

Reply: We have modified the sentence.

Line 90-91: Please revise the sentence.

Reply: We have revised the sentence.

Line 94-95: Instead of “breastfeeding period” please revise it to read “the first year of life”.

Reply: Thank you; we have changed the sentence.

Line 99: It will be appropriate to define the terms exclusive breastfeeding and partial breastfeeding as used in the study.

Reply: We have added a definition of breastfeeding.

Line 102: How was the breastfeeding satisfaction analyzed using the visual analogue scale?

Reply: We used a Mann-Whitney U-test. We have clarified the analysis section.

Line 105-106, 111-112: What is the difference between the two questions stated? 

Reply: It is the same question. We wanted to describe that we collected qualitative data from the free text alternative on the stated question.

Line 111: Please clarify which set of questions were asked at the different points. 

Reply: We have revised the measure section.

Results

Line 117-118: Please revise this sentence for clarity.

Reply: We have revised the sentence.

Line 124: Was this the only condition for conducting a Mann -Whitney U test analyses? You need to assess the data to see if they meet the assumptions for parametric testing or not before selecting the appropriate statistical test.

Reply: We have revised the sentence.

Line 125: Chi-square test is used to measure association between two categorical variables and not for measurement of differences.

Reply: We have revised the sentence.

Line 129: Only one open ended question (line 109-110), has been stated in the manuscript. What was the second open ended question?

Reply: It was a free text option on the question “If you have ceased breastfeeding, did you breastfeed as long as you wanted”. As stated in the measurement section under qualitative data. We have also revised the paragraph.

Line 134-137: There is a need to clearly define each of the alphabets or letters being used to describe participants and breastfeeding cessation for readers to understand.

Reply: We have revised the sentence.

Line 138-140: Please revise the statement for clarity. 

Reply: We have revised the sentence.

Line 149-150: For the written comments about breastfeeding cessation did participants provide more than one specific comment? (165 comments from 149 participants) 

Reply: Yes they did.

Table 1. The numbers in the table do not add up to the total number of participants

Reply: Thank you, we have corrected the numbers.

What was response rate in terms of the questionnaire sent out. Where the questionnaires self-administered?

Reply: We have added a flowchart over included and excluded participants and response rate to the questionnaires (figure 1). The questionnaires were self-assessed.

Table 1: “Partial breastfeeding 12 months after birth”. This percentage seems inaccurate, what was the denominator used for this calculation?

Reply: Thank you. There were missing data on 42 mothers, we have added this information in the table.

Table 2: Low maternal educational level…These numbers in the brackets do not sum up to 100%. What does this mean? If the numbers that responded to each question item are not same, you can put the total numbers beside the item.

Reply: Thank you for pointing out table 2. We have added data on both groups in each analysis to make it easier to interpret and to see associations with cessation of breastfeeding and if the mothers breastfeed as long as they wanted or not.

Were multiparas involved in this study? If yes, were there any differences found in breastfeeding cessation between primiparas and multiparas?

Reply: Both primipara and multipara mothers participated in the study. There was a difference in exclusive breastfeeding at discharge between primipara and muiltipara mothers participating in this study where more multipara mothers breastfeed exclusively. There were no statistically difference in the later follow-ups.

If different categories of preterm infants were involved in the study, were there any differences in terms of breastfeeding cessation?

Reply: There was no differences in cessation of breastfeeding at any of the follow-ups between infants born <32 or 32-36 weeks of gestational age participating in this study.

Line 172: Two open ended questions meanwhile there is only one in the manuscript.

Reply: There is two questions asked where the mothers could wright in free text, we have clarified the section in the measure section.

Line 174: Please reframe this sentence to clarify the main theme.

Reply: We have revised the sentence.

Line 179: “Negative feedback”. What exactly does this mean?

Reply: We have revised the sentence.

Line 182-183: If participant stopped breastfeeding at 3 months why the use of letter ‘p’ at 3 months. Please clarify.

Reply: Thank you for noticing that, it was a typo that we have corrected.

Line: 185 -186: Please move this statement upwards, i.e., after the first two statements and before the statement beginning with "From the mothers’ descriptions ..." (line 177)

Reply: We have moved the statement.

Line 189: “There was a difference in this theme” This is unclear, please revise

Reply: We have revised the sentence.

Line 219: Please be specific, “this pain was because the infant teeth came in”. Do you mean the infants with teeth were biting on the breast? 

Reply: We have revised the sentence.

Line 220- 221: “In addition, the infant health such as cowmilk protein allergy…” This statement seems unclear please consider revising.

Reply: We have revised the sentence.

Line 224: If participant ceased breastfeeding at 8 months why is the same participant categorized as practicing exclusive breastfeeding at 12 months?

Reply: Thank you for noticing that, it was a typo that we have corrected.

Line 246: Why was the quote of the participant referenced as [16]?

Reply: Thank you for noticing that, we have removed the reference.

Line 260: Mothers who did not breastfeed as long as they wanted were “less active …….” Did you observe this or this is an information they told you? Please clarify.

Reply: We observed that. We have revised the sentence.

Line 248- 250: The sentence looks incomplete.

Reply: We have revised the sentence.

Line 263 What is the meaning of “harmonious breastfeeding”?

Reply: We have clarified the sentence.

Line 264: How powerful can these words be when they are describing their experience?

Reply: We have revised the sentence.

Line 266: ……. “of pre-determined biologically given end “this section seems unclear, please revise.

Reply: We have revised the sentence.

Line 270: What does it mean for the mother to be “unreflective in relation to her own body”?

Reply: We have revised the sentence.

Discussion

Line 306: You need to cite more than one reference as you made mention of “studies”

Line 307: Same comment as line 306 above

Reply: We have added references.

Line 314: “The mothers did not seem to do anything active about their milk supply” What does this mean and what were the expectations in such situations?

Reply: We have clarified the sentence. 

Line 318: “Cessation of breastfeeding is a fact” revise this phrase.

Reply: We have revised the phrase.

Line 332- 333: “……the cultural influence and that mothers seem……” This information was not reflected in your results.

Reply: We have removed the information.

Line 334: In what ways can mothers be strengthened?

Reply: We have clarified the sentence.

Line 343: Approximation of 35 weeks of breastfeeding to 12 months needs to be clarified.

Reply: We have clarified the sentence. 

Line 345: ….. “when it was cultural acceptable to cease breastfeeding, this information did not reflect in your results.

Reply: We have removed the sentence.

Line 346: To what extent is breastfeeding for 2-4 years in non-industrial populations comparable to Sweden? Please expand the point for clarity especially with respect to the early cessation of breastfeeding reported in Sweden.

Reply: We have added two sentences for clarity. 

Line 348-349: Hypothesis? Please provide references for the statements made.

Reply: We have removed the hypothesis.

Line 354: “A limitation is that comments left by mothers were relatively short”. If this is addressed as a limitation, how does the qualitative data adds up to the strength of your study as earlier described in line 352?

Reply: We have revised the sentence.

Line 355: “width and depth”, What does this mean in the context of your study?

Reply: We have revised the limitation section.

References: All references used were accounted for in the list of references, however there is a need to cross check reference number [16] since it was used as in text citation for a quote by a participant (Line 246).

Reply: We have removed the reference.

---

## [Editor Report · Decision Letter 1]

30 Apr 2020

Cessation of breastfeeding in mothers of preterm infants – a mixed method study

PONE-D-20-00937R1

Dear Dr. Ericson,

We are pleased to inform you that your manuscript has been judged scientifically suitable for publication and will be formally accepted for publication once it complies with all outstanding technical requirements.

With kind regards,

Joann M. McDermid, MSc, PhD, RDN, FAND

Academic Editor

PLOS ONE
---

## [Editor Report · Acceptance letter]

5 May 2020

PONE-D-20-00937R1 

Cessation of breastfeeding in mothers of preterm infants – a mixed method study 

Dear Dr. Ericson:

I am pleased to inform you that your manuscript has been deemed suitable for publication in PLOS ONE. Congratulations! Your manuscript is now with our production department. 

With kind regards,

on behalf of

Professor Joann M. McDermid 

Academic Editor

PLOS ONE